# Brazilian and Mexican propolis and their possible mechanism of action against non-enveloped viruses

Norma Patricia Silva-Beltrán[1,2*], Lenin Domínguez-Ramírez[3], Stephanie A. Boone[1*], Charles P. Gerba[1], Luis Alberto Cira-Chávez[4], M. Khalid Ijaz[5], Julie Mckinney[5]

1 Department of Environmental Science, university of Arizona, Water Energy Sustainable Technology (WEST) Center, Tucson, Arizona, United States of America, 2 Departamento de Ciencias de la Salud. Universidad de Sonora, Campus Cajeme, Ejido Providencia, Cd. Obregón, Sonora, México, 3 Centro de Investigación Biomédica de Oriente(CIBIOR), Instituto Mexicano del Seguro Social (IMSS), Atlixco, Mexico, 4 Departamento de Biotecnología y Ciencias Alimentarias. Instituto Tecnológico de Sonora. Cajeme, Sonora, México, 5 Global Research and Development for Lysol and Dettol, Reckitt Benckiser LLC, Montvale, New Jersey, United States of America

* Sboone@arizona.edu (SAB); normasilva@arizona.edu (NPSB)

## Abstract

Propolis is a resinous substance collected by honeybees and is mostly composed of polyphenols which vary by geographical location. This study investigated the possible mechanism of action of phenolic compounds of Brazilian and Mexican green and red propolis against two non-enveloped viruses. Bacteriophage surrogates, ΦX174 and MS2 were used to assess antiviral properties. Propolis samples were characterized by performing a phenolic profile using ultra-performance liquid chromatography (UPLC), which included 12 phenolic compounds such as phenylpropanoids, flavonoids, phenols, and phenolic. Quercetin, eugenol, kaempferol and naringenin were the most abundant compounds found in propolis. *In silico* molecular docking was also conducted to determine binding energy and molecular interaction and putative mechanism of propolis phenolic compounds with two viral capsid proteins and two proteins involved in viral replication and infection. The best antiviral effect was in green propolis with a $\sim 3,1$ and $\sim 4.5 \log_{10}$ reduction in MS2, and ΦX174, respectively. Molecular docking simulations revealed that ΦX174 was also more sensitive to the phenolic compounds and that the combination of quercetin and kaempferol showed the greatest antiviral effect as a possible mechanism, through binding to the viral capsid proteins near the viral genome binding sites.

## Introduction

Propolis are resin like products produced by honeybees (*Apis mellifera*) with substances collected from different plants and mixed with a β-glucosidase from the bees' saliva. After being digested by the bees, these substances are incorporated into the

**Data availability statement:** If the data are all contained within the manuscript and/or Supporting Information files, enter the following: All relevant data are within the manuscript and its Supporting Information files.

**Funding:** The author(s) received no specific funding for this work.

**Competing interests:** The authors have declared that no competing interests exist

honeycomb as a protect mechanism [1]. Propolis has been used as a medication in human disease treatment because of its antioxidant, anti-inflammatory and antimicrobial activity. For instance, since the 17th century, London pharmacopeias have included propolis as an official drug [2]. Recent studies with a Bulgarian propolis used against structurally different human viruses (HCoV OC43, HRSV-2, HSV-1, HRV-14, HadV5) showed positive results in terms of reducing viral infectivity on enveloped viruses [3]. Some reports on propolis from Mexico and Brazil also have shown antiviral efficacy [4–7].

Many compounds are present in propolis, including over 800 chemical fractions [1,8]. Their composition depends on the specific properties of geographic regions where these types of propolis are found [1]. Mexican and Brazilian propolis are found in different hemispheres, and some reports show that although these propolis come from very diverse ecosystems they share similar compounds, for instance phenolic acids, phenolic acid esters, flavonoids, terpenoids trans ferulic acid, caffeic acid, quercetin, apigenin, kaempferol, naringenin among others are present in both Mexican and Brazilian propolis and contain biological activity [9]. The mechanism of antiviral activities of the different individual compounds present in propolis have been evaluated using molecular docking [10]. Molecular docking is a computer modeling tool recently used to understand the biomolecular interactions of the chemicals present in propolis. The model predicts the preferred orientation of a compound (ligand) in the ideal binding site of the specific target area of the protein (receptor) thus forming a stable complex with free energy that can be estimated [11]. It can also provide insight to further understand their mechanism of action as well as ways to improve their antimicrobial action through changes in formulation.

Several studies have employed bacteriophage models to investigate the in vitro virucidal effects of natural substances [4,12]. Among them, the bacteriophages MS2 (single-stranded RNA) and ΦX174 (single-stranded DNA), both non-enveloped viruses, have been employed as viral models. Non-enveloped viruses are typically more resistant to adverse environmental conditions and to the action of antimicrobials [13]. The MS2 bacteriophage has been used as an experimental surrogate for SARS-CoV-2 [14], while ΦX174 used as an enteric virus surrogate [15]

In this context, the present work evaluates the antiviral effect of Brazilian and Mexican propolis using two non-enveloped viruses. Furthermore, through molecular docking, the possible mechanism(s) of action can be identified by evaluating the interactions of the different chemical compounds present in propolis against two capsid proteins and two proteins involved in for MS2 and ΦX174 replication.

## Materials and methods

### Reagents

Standards, gallic acid (3,4,5-trihydroxybenzoic acid), resorcinol, ferulic acid, rutin (Quercetin 3-O-rutinoside), naringenin (5,7-dihydroxy-(2–4-hydroxyphenyl) chroman-4-ona), vanillin (4-Hidroxy-3-methoxybenzaldehyde), kaempferol, eugenol (4-Allyl-2-methoyphenol), quercetin, chlorogenic acid, coumaric acid, ferulic acid and caffeic acid were obtained from Sigma-Aldrich (St. Louis, MO, USA). Water,

methanol, acetic acid, acetonitrile (high-performance liquid chromatography (HPLC) grade and dimethyl sulfoxide (DMSO) were obtained from J.T. Baker (Baker-Mallinckrodt, Mexico City, Mexico).

## Bacterial host and bacteriophages

The host bacteria *Escherichia coli* (13706-B1) and *E. coli* (15597-B1) for Φ-X174 and MS2, respectively were obtained from the American Type Culture Collection (Manassas, VA).

## Propolis

Green and red propolis samples from Alagoas (Northeast region), and the second red propolis samples from Salvador, Bahia (Northeast region) were obtained directly from collectors in 2020. Green propolis sample from Sonora state (Northwest region) was a donation from Martín Guadalupe Rodríguez of Cooperativa mieles de Cajeme.

**Obtaining propolis ethanolic extracts.** Extracts of different origin were prepared according to Sokolonski et al. [16]. Propolis (2 g) were dissolved in 15 mL of 80% ethanol by mixing the samples for 30 minutes under constant agitation to obtain propolis extracts (G, RA, RB, and GS) (**Table 1**). The extracts were centrifuged (Beckman, Model J2-21, USA) at 10,000 RPM for 15 min at 4°C. The supernatants were subsequently dried at 45 °C in a hood with air circulation until no further weight loss was observed, indicating that the constant weight had been reached. Once constant weight was achieved, the extracts were weighed and then diluted according to the study.

## Viral propagation

Single-host bacterial colonies from overnight cultures on trypticase agar (TA) were cultured in 10 mL of trypticase soy broth (TSB) at 37°C for 24 h until they reached the exponential growth phase. After incubation, 1 mL of bacteria was added to 100 mL of TSB under constant agitation for 4 h to achieve log phase. Subsequently it was mixed with 100 µL of a bacteriophage suspension into 4 mL of TSB containing 0.8% agarose and 0.5 mL of host bacteria. This mixture was placed onto TSA Petri dishes and incubated for 24 h at 37°C. After incubation, 6 mL of buffer solution was added, and the Petri dishes were oscillated in the multi-purpose rotator for 3 h. The surface layer of each Petri dish was recovered by removal with a bacteriological loop, and the resulting suspensions were placed in sterile tubes. The samples were twice centrifuged at 10,000 × g for 20 min at 4 °C, and the supernatants were filtered through a cellulose acetate membrane (Whatman, USA) with a pore size of 0.2 µm.

## Antiviral assay

Serial dilutions of extracts (G, RA, RB, GS) were prepared in 1% dimethyl sulfoxide (DMSO), and filtered using a 0.22 µm syringe filter. They were exposed to 500 µg/mL of extract by addition of 100 µL of phage suspension into 5 mL of extract. Two contact times were evaluated: 10 and 30 min. Next, serial dilutions of $10^{-2}$ to $10^{-10}$ were made in phosphate buffered saline (PBS). To tubes containing 5 mL of 0.8% TSB agarose, 100 µL of each dilution and 500 µL of the host cell in log phase were added. The mixtures were poured into Petri dishes containing solidified trypticase soy agar (TSA), and incubated at 37 °C for 24 h, and the plaque-forming units (PFU) per milliliter determined. Controls were in PBS without extract were also performed.

**Table 1. Origin and nomenclature of propolis samples.**

| Sample | Origin | Nomenclature |
|---|---|---|
| Green Propolis | Alagoas (Brazil) | G |
| Red Propolis | Alagoas (Brazil) | RA |
| Red Propolis | Salvador, Bahía (Brazil) | RB |
| Green Propolis | Sonora (Mexico) | GS |

## UPLC analysis

The chromatographic analysis was performed according to Balderrama-Carmona et al. [17] in an UPLC equipped with a diode array UV detector (Walters, Milford, MA). An Acquity UPLC BEH C18 1.7 µm column (2.1 x 50 mm) was used. The phenolic compounds were quantified using external standard curves prepared from pure standards. For compound identification, the absorption spectrum generated by integrating the area under the curve at the retention time detected in the standards was compared. Quality parameters were reproducibility, linearity, and limit of quantification (LOQ). Likewise, three mobile phases were used to promote the separation of the phenolic compounds: (A) water with 0.1% acetic acid; (B) methanol; and (C) acetonitrile, all solvents were with HPLC grade; with a total running time of 14 min at a temperature of 35 and 20 °C for column and sample, respectively, at a wavelength of 280 nm. An elution gradient was used starting with 90% (A), 5% (B) and 5% (C); changing the ratio at 6 min with 76% (A), 12% (B) and 12% (C); at 11 min with 36% (A), 32% (B) and 32% (C); finally, changing to the initial gradient at 12 min until the run was finished.

## Molecular docking

Docking to the bacteriophage MS2 capsid protein was done using PDBID 1AQ3 [18] while bacteriophage ΦX174 was done using PDBID 2BPA [19]. For docking against spike proteins, we used 5TC1 [20] chain M for MS2 and 1 CD3 [21] for ΦX174 chain G. Water molecules and nucleotides were removed before docking. The binding site was selected based on the polynucleotide binding site of each capsid protein or spike protein as identified from the deposited structures. Vina 1.2.3 [22] (https://autodock-vina.readthedocs.io/en/latest/docking_multiple_ligands.html) was used as it allows the use of two ligands simultaneous binding events. That is, in Fig 1A, the label "Quercetin" in the X-axis indicates that docking took place by using" Quercetin" and each of the other ligands (see **Table 2**). Exhaustiveness was set to 100 and ran on a 6-core computer. Previous reports put Vina average AUC at 0.71±0.12, and a 68% success rate when considering the top pose only. Visualization was performed using UCSF Chimera [23].

## Data analysis

The statistical analysis used to evaluate the survival of virus was random and considered two factors. The factors were as follows: samples G, RA, RB, GS; contact time were the other factor that is 1, 10 min. Considering the variability of the response, the reduction was scored as a percent of $\log_{10}$ reduction. The treatments were performed in triplicate. Statgraphic plus 5.1 was used to perform the ANOVA.

## Results

### Phenolic composition of propolis

A phenolic profile was performed on samples G, GS, RA, and RB using 12 standards. Standard curves for the phenolic compounds are shown in S1 Fig, while chromatograms and detection peaks for G, GS, RB, and RA are provided in the supplementary material (S2, S3, S4, and S5 Figs). Quality control parameters included linearity ($R^2 > 0.99$) and precision (RSD < 4.5%), as detailed in S1 Table. Linearity was evaluated by analyzing different standard solutions, and calibration curves were constructed by plotting the under-curve area versus concentration. The curves exhibited slopes of 86.136, intercepts of -105.74, and correlation coefficients of 0.99, with a limit of quantification (LOQ) of 0.2 µg/mL.

A total of 10 constituents were identified, and **Table 2** shows the concentration of the phenolic components detected in the red propolis from Brazil, as well as in the green propolis from Brazil and Mexico (Table 2).

Results are expressed as mean±standard deviation (n=3) and correspond to the milligrams per gram of dry weight. ND: Not detected. G Green propolis Alagoas Brazil; RA Red propolis Alagoas Brazil; RB Red propolis Salvador Bahia Brazil; GS Green propolis Sonora, Mexico.

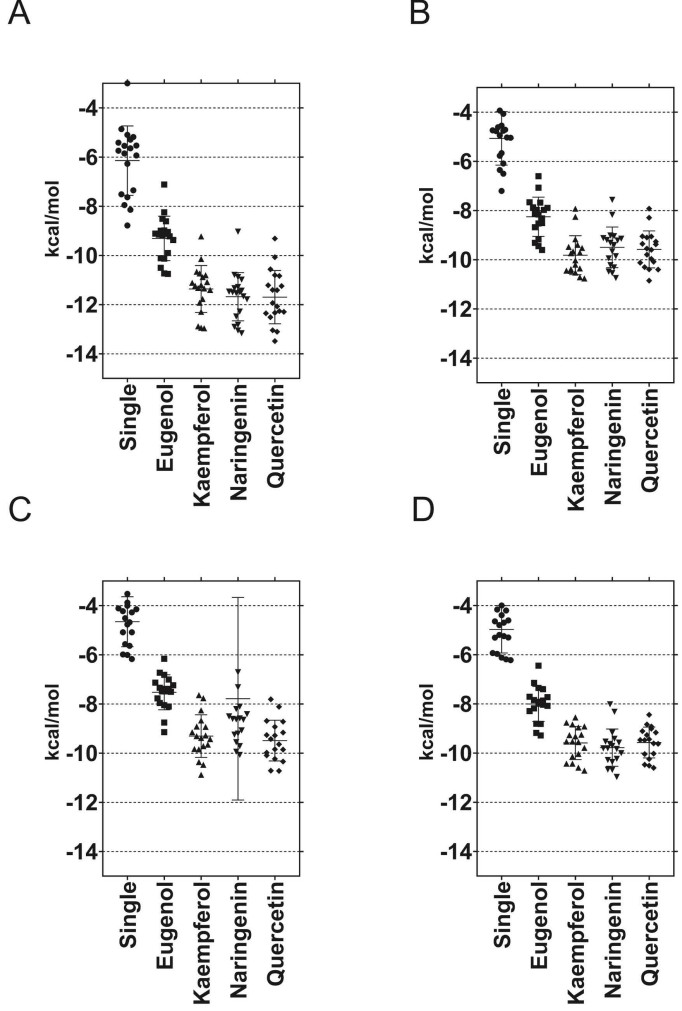

**Fig 1. Molecular docking results for the different viral targets. Docking was performed against a single ligand (indicated in the x-axis as Single) or against two ligands keeping one fixed. See methods. Panels A and C correspond to capsid proteins while panels B and D correspond to infectives proteins. A and B are proteins from ΦX174 while C and D belong to MS2, respectively.**

It was observed that green propolis of both countries has a greater number of constituents, in the case of G, the three main compounds detected were in the following order of abundance; Coumaric acid> eugenol> kaempferol for GS, eugenol >caffeic acid >kaempferol, in the case of RB: Quercetin> kaempferol >naringenin, and RA: quercetin> kaempferol > eugenol. In all propolis the presence of quercetin was detected, in the case of both red propolis a higher concentration of these constituents was observed in a concentration of 71.21 mg/g for RB. Although other reports in Sonora propolis have shown the presence of rutin and resorcinol [24], but in our samples, it was not detected. Likewise, it was observed that the most abundant constituent of GS was caffeic acid at a concentration of 14.26 mg/g and for G it was coumaric acid at a concentration of 60.94 mg/g.

## Antiviral effect of different concentrations and contact time of propolis on MS2 and ΦX174

The antiviral activity of the different types of propolis (G, GS, RB, and RA) was determined by monitoring the reduction of the plaque-forming unit (PFU) after 10 and 30 minutes of exposure compared to controls. **Table 3** shows the antiviral

**Table 2. Identification and quantification of the phenolics constituents of propolis samples expressed in mg/g propolis.**

| Phenolic compounds | G | GS | RB | RA |
|---|---|---|---|---|
| | mg/g propolis | | | |
| Gallic acid | ND | 0.114 ± 0.002 | ND | ND |
| (3,4,5-trihydroxybenzoic acid) Resorcinol | ND | ND | ND | ND |
| Chlorogenic acid | 16.127 ± 0.971 | ND | ND | ND |
| Caffeic acid | ND | 14.26 ± 0.744 | ND | ND |
| Vanillin (4-Hidroxy-3-methoxybenzaldehyde) | 1.285 ± 0.005 | 1.671 ± 0.043 | ND | ND |
| Coumaric acid | 60.949 ± 1.349 | 0.154 ± 0.005 | ND | ND |
| Ferulic acid | ND | 0.246 ± 0.013 | ND | ND |
| Rutin (Quercetin 3-O-rutinoside) | ND | ND | ND | ND |
| Quercetin | 6.933 ± 0.897 | 0.117 ± 0.004 | 71.210 ± 3.340 | 24.260 ± 1.902 |
| Naringenin(5,7-dihydroxy-(2–4-hydroxyphenyl) chroman-4-ona) | ND | 6.248 ± 0.134 | 27.766 ± 0.112 | 0.9797 ± 0.003 |
| Kaempferol | 45.752 ± 2.875 | 10.077 ± 0.359 | 73.869 ± 0.511 | 38.470 ± 0.266 |
| Eugenol (4-Allyl-2-methoyphenol) | 56.570 ± 4.985 | 27.496 ± 1.320 | 2.060 ± 0.079 | 1.071 ± 0.004 |

**Table 3. The reduction ($\log_{10}$ PFU/mL) in bacteriophages MS2, and ΦX174 for G, GS, RB, RA in 500 µg/ml.**

| Propolis sample | Time (1min) | MS2 | Φ- X174 |
|---|---|---|---|
| | | Reduction ($\log_{10}$PFU/mL) | |
| G | 10 | 2.0988 ± 0.0812 | 3.0341 ± 0.0330 |
| | 30 | 3.4587 ± 0.0602 | 4.5800 ± 0.5443 |
| GS | 10 | 2.0383 ± 0.4194 | 3.1729 ± 0.1125 |
| | 30 | 3.1866 ± 0.2758 | 4.1008 ± 0.0211 |
| RB | 10 | 1.2996 ± 0.0832 | 3.0939 ± 0.3169 |
| | 30 | 2.1617 ± 0.4312 | 3.2438 ± 0.1368 |
| RA | 10 | 1.0100 ± 1.0244 | 3.0378 ± 0.0314 |
| | 30 | 2.4170 ± 1.1821 | 3.8561 ± 0.0414 |

All values represent mean of triplicate determinations ± SD

activities of the extracts (G, GS, RB, and RA). The antiviral activity in the four propolis was confirmed by significance reduction in PFU/ml. Green propolis G, and GS reduced both viruses by ~ 2.5 $\log_{10}$ in 10 minutes and 4.5 $\log_{10}$ in 30 minutes.

Red propolis RB and RA reduced both viruses by ~ 1 $\log_{10}$, however at 30 minutes both RB this increased up to 3 $\log_{10}$.

In general, for all treatments it was observed that MS2, an RNA genome, is less sensitive to the effects of the propolis evaluated than the DNA genome virus Φ- X174.

## Molecular docking

Molecular docking is a useful technique to model protein-ligand interactions based on known structures, usually obtained by crystallographic, NMR or electron microscopy methods. To assess the antiviral effect of compounds identified as components of propolis, we docked them to capsid and spike proteins for both viruses used in this study. A first run, docking each ligand alone, showed a wide range of affinities from very low (-3 kcal/mol) to medium affinities (-8.7, **Fig 1**, results labeled as "Single").

To test synergy between compounds different combinations of phenolics compounds detected in G, GS, RB and RA, we carried out docking using the two ligand-method. Using this method, similar to a two ligand enzyme kinetics, one

**Table 4. Docking affinities (kcal/mol) for pair of phenolic compounds bound near the nucleic acid binding site on ΦX174 capsid.**

| | Chlorogenic acid | Kaempferol | Quercetin |
|---|---|---|---|
| Chlorogenic acid | | | −11 |
| Kaempferol | | −12.4 | −11.6 to −13.0 |
| Quercetin | −11 | −11.6 to −13.0 | −12.2 |

ligand is always the same (i.e., quercetin) and the other is variable (all from the list in **Table 2**, including eugenol). The result is the sum of the affinities of both ligands. For our four docking targets, docking two ligands increases the affinity. Ligands selected as fixed were quercetin, naringenin, kaempferol and eugenol because they occur in the majority of the propolis evaluated in the current work. It is the case of the ΦX174 capsid where greater affinity is observed, reaching a maximum of -13.48 kcal/mol when docking quercetin+quercetin, -13.16 kcal/mol for naringenin+quercetin, and -12.96 kcal/mol for kampferol+kampferol (see **Table 4**). In contrast, MS2 capsid displayed few affinities below -10 kcal/mol. The infections proteins for ΦX174 and MS2 behave similarly with low affinities. With this in mind, we focused on the analysis of ΦX174 and MS2 capsid results.

To select those pairs of compounds that might affect the interaction between the viral capsid and nucleic acid, we visualized the results for ΦX174 capsid. Many results are compounds bound at two different regions of the protein surface; just a few are near one another as well as close to the ssDNA binding site (**Figs 2A to C** and **Table 4**). The combination of chlorogenic acid+quercetin is unique to G propolis. From the results, it is difficult to estimate if the phenolic compound would be competitive with ssDNA binding or if binding could lead to a conformational change affecting ssDNA binding. This latter possibility is particularly relevant for most pair of compounds that were bound at sites distant from that of ssDNA (shown in **Fig 2D**).

In the case of MS2 capsid results, it is apparent that binding of phenolic compounds are binding in the same surface as the tip of the ssRNA molecule solved in the structure. Their affinities are about mean 2 kcal/mol lower that the results of ΦX174 but their position suggest competitive binding (**Fig 3**). For this target, the combination coumaric acid+quercetin is particular to G propolis.

In general terms, the individual compounds in all the extracts show poor affinities for the viral capsid proteins, on the contrary when they are combined the binding energy increases and in this study it was observed that the best binding affinity that was expressed in the capsid of the ΦX174 virus, suggesting that the phenolic compounds detected in this study have better antiviral activity in the DNA virus, and these results coincide with those observed in the *in vitro* study; it is also apparent that GS propolis would target ΦX174 better and G propolis would target MS2. This conclusion can be drawn from **Table 2** (propolis composition) and **Tables 4** and **5** (compound pair affinities). The other combinations with high affinity (kaempferol and quercetin) are common to the four propolis (G, GS, RA, and RB).

In reference to the results for the ssRNA-coat protein (chain M) responsible for attaching the virus to an F-pilus bacteria [20] and ssDNA-spike protein (chain G) which bind to LPS bacteria [21], they are more difficult to interpret because they present many different regions of union and show less binding affinities to the compounds detected in G, GS, RA and RB, suggesting that these propolis could bind to these proteins with a unknown effect.

## Discussion

Propolis extracts from Mexico and Brazil were analyzed to evaluate their antiviral activities *in vitro* against non-enveloped virus and to explore possible mechanisms using molecular docking on the detected chemical constituents. A targeted phenolic profile was performed on samples G, GS, RA, and RB, including phenylpropanoids or phenolic acids (caffeic acid, ferulic acid, and p-coumaric acid), flavonoids and their derivatives (rutin, quercetin, naringenin, kaempferol), phenols (resorcinol, eugenol, vanillin), and phenolic esters such as chlorogenic acid and gallic acid, as detailed in **Table 2**. The presence of these compounds was corroborated through chromatographic analysis.

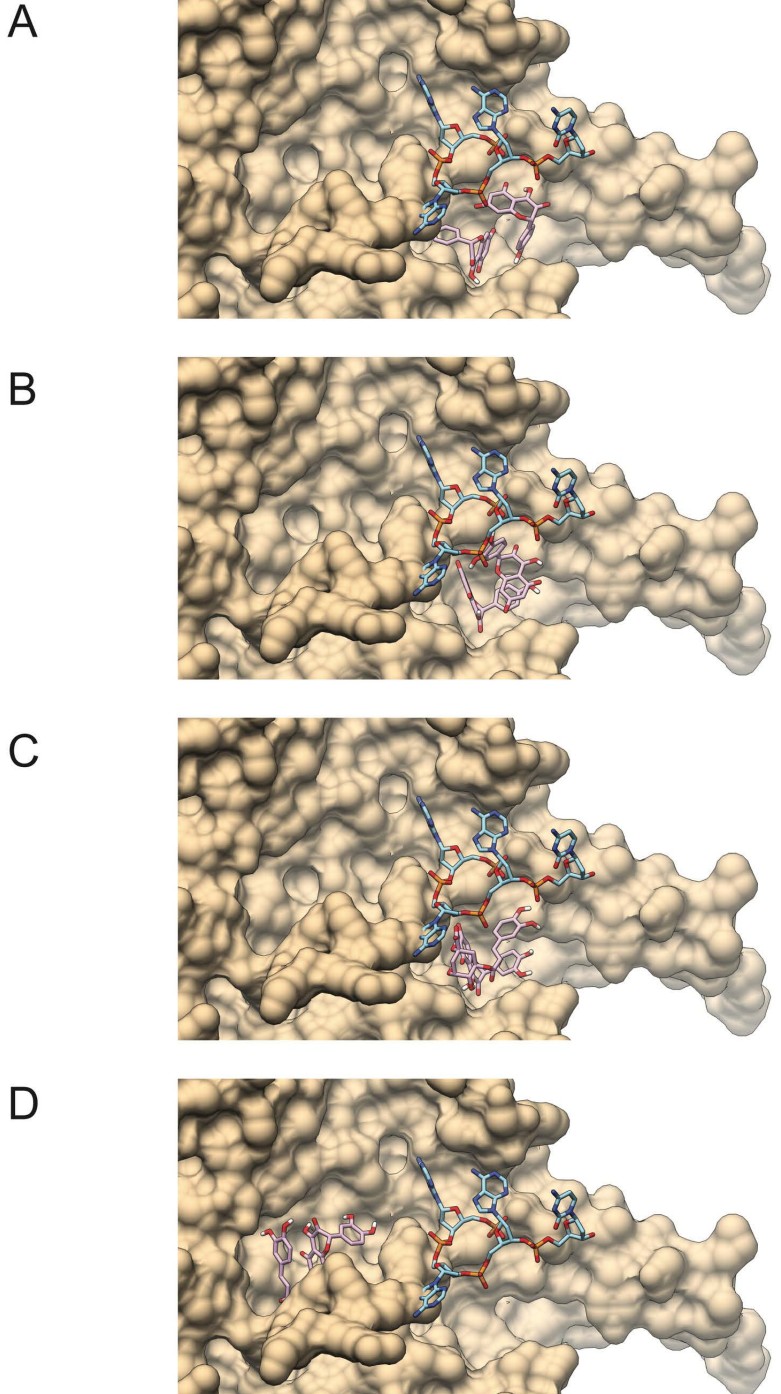

**Fig 2. Two-ligand docking visualization for ΦX174 capsid protein surface and the ssDNA binding site. Pair of ligands show in sticks are kaempferol+quercetin (A), kaempferol+**kaepmferol (B), **and quercetin+chlorogenic acid (C). In D we show quercetin+chlorogenic acid in a secondary binding site away from the ssDNA. The capsid protein surface is show in teal. ssDNA strand is show but was not present during docking.**

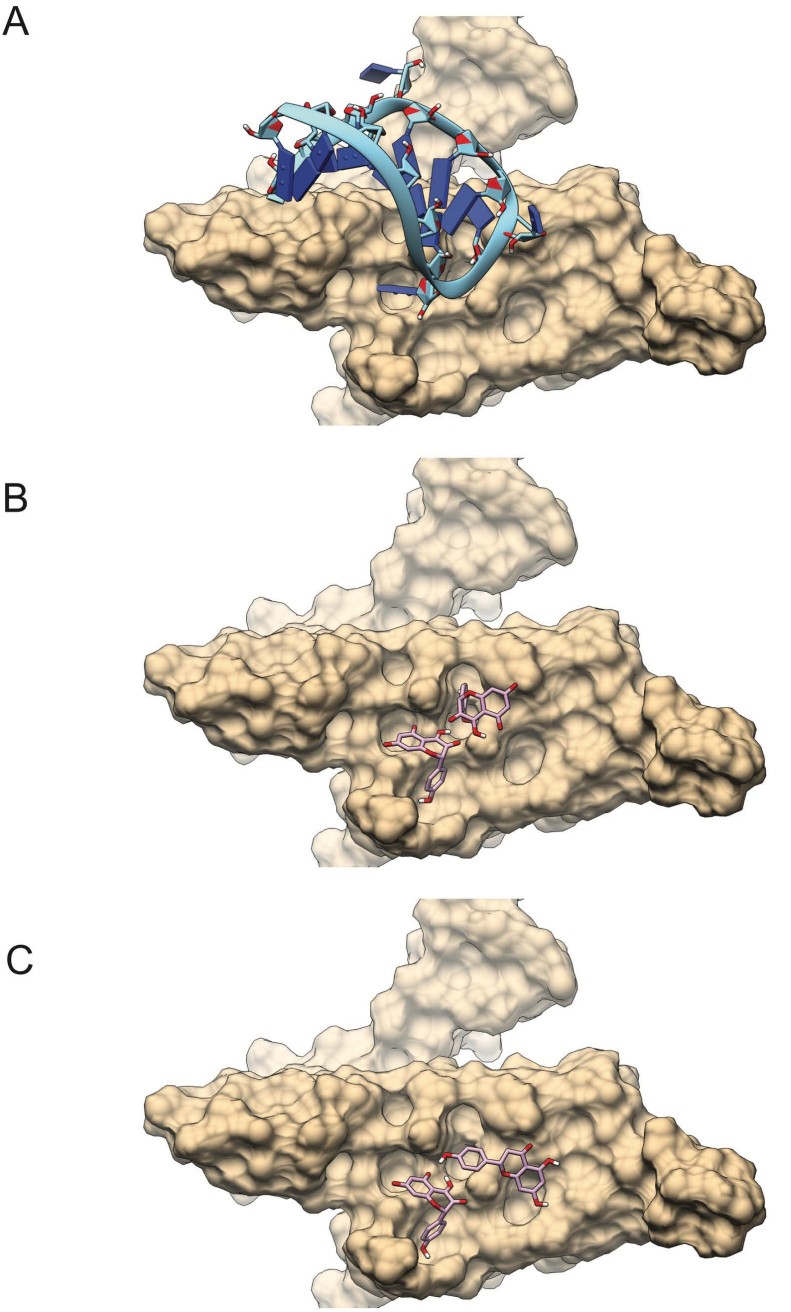

**Fig 3. Two-ligand docking visualization for MS2 capsid protein surface and the ssRNA binding site. The hairpin RNA is show nested on the surface of the protein with bases in direct contact with the protein(A), quercetin+kaempferol (B), and naringenin+kaempferol (C). The capsid protein surface is show in teal while the ssRNA strand is show but was not present during docking.**

The phenolic composition of propolis from the state of Sonora Mexico has been previously studied by Valencia et al. [25]. Mexican propolis are derived from Sonoran propolis mainly from *Encelia farinosa*, which is a shrub identified of semi-arid areas [26]. Its main components are pinocembrin, pinobanksin and CAPE (caffeic acid phenethyl ester), caffeic acid and various flavonoids. In the present study, caffeic acid was detected at a concentration of 14.26 mg/g, representing

**Table 5. Docking affinities (kcal/mol) for pair of phenolic compounds bound near the nucleic acid binding site on MS2 capsid.**

|  | Coumaric acid | Kaempferol | Quercetin |
|---|---|---|---|
| Coumaric acid |  |  | −10.2 |
| Kaempferol |  | −10.8 | −10.3 to −10.7 |
| Quercetin | −10.2 | −10.3 to −10.7 | −10.7 |

a significant concentration as reported as noted in propolis from southern Mexico previously [27]. However, this phenyl-propanoid was only detected in GS. Brazilian green propolis is mostly composed of prenylated phenylpropanoids, mainly artepillin C; Brazilian red propolis mainly contains isoflavonoids (biochanins, medicarpin) [28]. Although we acknowledge the potential presence of other characteristic compounds in the studied propolis samples, this study focused on identifying compounds that, based on previous research conducted by our team, we knew could be detected in both Brazilian and Mexican propolis [9]. This approach ensured the validity of the docking analysis. Consequently, were also detected in Brazilian propolis compounds, such as chlorogenic acid, caffeic acid, vanillin, coumaric acid, ferulic acid, naringenin, kaempferol, eugenol and quercetin [9]. Quercetin, kaempferol and eugenol were detected in all propolis samples G, GS, RB, RA. In addition to the compound identified, it was observed that red propolis presents significant concentrations of kaempferol with 45.752, 73.86 and 38.47 mg/g for G, RB, RA respectively. However, rutin (quercetin 3-O-rutinoside) was not detected in all propolis samples, likely because the flavonoids in propolis are mostly aglycons, making rutin a rare major component. Another compound that was not detected was resorcinol, which was expected to be found in Brazilian propolis, particularly in samples from hives near mangroves in the states of Bahía (RB) and Alagoas (RA) [29]. The absence of specific compounds in propolis depends on the botanical diversity of the region.

Both propolis from the Northeast and Northwest of Brazil and Mexico, respectively, were evaluated for their antiviral capacity against two non-enveloped viruses. G, GS, RB and RA were effective against MS2, linear ssRNA, and ΦX174, linear ssDNA (**Table 3**). These results support previous work carried out by our team on Brazilian and Mexican propolis where it was also showed that propolis from Brazil (Southeast, South regions), and Mexico (Northwest region) resulted in the reduction of viruses against the ssRNA coronaviruses. In addition, it was also observed that the antiviral efficacy of propolis depending on their region of origin and treatment of the extract [4,7]. This research evaluated propolis from Northeast region from two countries: Brazil and Mexico. The results showed viral reductions between 2 and 3 $\log_{10}$ after 30 minutes of contact time for the RNA virus, while for the DNA virus the reductions were between 3 and 4 $\log_{10}$. The highest efficacies were shown in green propolis G and GS. This matches the results obtained from molecular docking.

We know that although these propolis come from very diverse ecosystems, they have a similar chemical composition, for instance, with phenolic compounds. Phenolic compounds are recognized as antiviral compounds and their mode of action includes multiple targets in the infectious cycle of the virus, such as preventing adsorption, viral attachment to the host cell, preventing its entry, inhibiting replication and assembly [13]. Most phenolic compounds come from natural products; several studies have observed that the antiviral effect of natural substances is greater against enveloped viruses [13] and that the combination of substances is more effective than the of any of the compounds separately, a phenomenon known as synergy; The synergistic effect of the phenolic antiviral molecules present in propolis has been studied *in vitro* against enveloped viruses such as herpes simplex and SARS-CoV-2 [30]. In present study, the synergistic activity of the phenolic compounds detected in G, GS, RB and RA against the surface proteins of two non-enveloped viruses, MS2 and ΦX174, were explored by molecular docking. RA and RB showed good binding affinity to the surface proteins of both viruses (**Fig 1**), however, the best energies were expressed in the 2BPA capsid proteins of the DNA virus ΦX174, which showed the best binding energy to the combination of polyphenolic compounds in G and GS. The single compounds alone expressed lower binding energy (**Fig 1**). This suggests that the combination of these components can weaken protein-nucleic acid interaction affecting negatively the viral infective cycle. Previous studies have showed that the *in vitro*

antiviral efficacy of propolis is more active when combinate compounds as flavonoids (kaempferol, and quercetin) [30]. Kaempferol and quercetin were flavonoids detected in high concentrations in G and GS. Several studies have shown that kaempferol and coumaric acid, both found in Brazilian propolis, can block surface proteins, thereby inhibiting viral entry. Additionally, kaempferol exhibited greater antiviral activity in both *in vivo* and *in vitro* studies compared to other structures, including coumaric acid. This hydroxycinnamic acid was also detected in G and GS [31,32]. Furthermore, **Fig 1A** shows that phenolic compounds exhibit greater affinity for DNA bacteriophages. This finding is consistent with the *in vitro* study (**Table 3**), which indicates that the DNA bacteriophage ΦX174 is more sensitive.

To link the *in vitro* effects with the known functions of specific viral proteins, we selected four proteins involved in capsid structure, genome binding, and infection mechanisms. Specifically, we analyzed MS2 (PDB ID: 1AQ3) and ΦX174 (PDB ID: 2BPA) both of which are required for genome attachment-dependent capsid assembly.

To determine the effect on viral processes, we selected spike proteins, including 5TC1 (chain M) for MS2 and 1 CD3 (chain G) for ΦX174. The former is responsible for virus attachment to the F pilus and genome delivery, while the latter is associated with effective infection [33]

Therefore, the antiviral properties of propolis G and GS observed in this study could be attributed to their higher content and composition of phenolic compounds, which enhance synergistic (protein-ligand) effects, facilitating interactions that neutralize the capsid proteins required for genome attachment-dependent capsid assembly of both non-enveloped viruses.

## Conclusion

The effect of propolis over bacteriophages ΦX174 and MS2 was investigated through a variety of wet-lab methodologies as well as *in silico* docking. Of the four propolis G, GS, RA and RB, originated in two different countries, two were the most successful: G and GS. However, all four share a chemical composition where quercetin+kaempferol shows high binding affinity to the ΦX174 and MS2 capsid proteins. Altogether, our results highlight the effect of propolis on phage infection and suggest a possible mechanism of action through binding to capsid proteins, which could prevent the proper formation of the capsid-genome complex and block viral infection.

## Supporting information

**S1 Table. Reproducibility test results.** For the retention times of 12 phenolic compounds analyzed using the described UPLC method.
(TIF)

**S1 Fig. UPLC chromatogram of phenolic compound standards analyzed at 280 nm [y axis = intensity (absorbance unit, AU); x axis = retention time (min).** Peaks: 1, gallic acid; 2, resorcinol; 3, chlorogenic acid; 4, caffeic acid; 5, vanillin; 6, coumaric acid; 7, ferulic acid; 8, rutin; 9, naringenin; 10, quercetin;11, kaempferol, and 12, eugenol.
(TIF)

**S2 Fig. UPLC chromatogram of phenolic compounds present in G.** Peaks: (3) naringenin, (6) coumaric acid, (10) quercetin, (11) kaempferol, (12) eugenol.
(TIF)

**S3 Fig. UPLC chromatogram of phenolic compounds present in GS.** (9) naringenin, (10) quercetin, (11) kaempferol, (12) eugenol.
(TIF)

**S4 Fig. UPLC chromatogram of phenolic compounds present in RB.** (3) naringenin, (4) caffeic acid, (5) vanillin, (6) coumaric acid, (7) ferulic acid, (9) naringenin, (10) quercetin, (11) kaempferol, (12) eugenol.
(TIF)

**S5 Fig. UPLC chromatogram of phenolic compounds present in RA.** (9) naringenin, (10) quercetin, (11) kaempferol, (12) eugenol.
(TIF)

## Acknowledgments

The authors wish to thank Mei-Li Portela for providing technical support, and Martín Guadalupe Rodríguez of Cooperativa mieles de Cajeme for the donations of propolis samples in Mexico.

## Author contributions

**Conceptualization:** Norma Patricia Silva Beltrán, Lenin Domínguez-Ramírez, Stephanie A. Boone, Charles P. Gerba.

**Formal analysis:** Norma Patricia Silva Beltrán, Lenin Domínguez-Ramírez.

**Investigation:** Norma Patricia Silva Beltrán, Lenin Domínguez-Ramírez, Stephanie A. Boone.

**Supervision:** Stephanie A. Boone, Charles P. Gerba.

**Writing – original draft:** Norma Patricia Silva Beltrán, Lenin Domínguez-Ramírez.

**Writing – review & editing:** Norma Patricia Silva Beltrán, Stephanie A. Boone, Charles P. Gerba, Luis Alberto Cira-Chavez, M. Khalid Ijaz, Julie McKinney.

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
