## [Decision Letter · Decision Letter 0]

22 Dec 2024

PONE-D-24-54321Brazilian and Mexican Propolis and their possible Mechanism of Action against non-enveloped VirusesPLOS ONE

Dear Dr. Beltrán,

We look forward to receiving your revised manuscript.

Kind regards,

José M. Alvarez-Suarez

Academic Editor

PLOS ONE

Journal Requirements:

2. Please upload a new copy of Figures S1, S2, S3, S4 and S5 as the detail is not clear. Please follow the link for more information: https://blogs.plos.org/plos/2019/06/looking-good-tips-for-creating-your-plos-figures-graphics/" https://blogs.plos.org/plos/2019/06/looking-good-tips-for-creating-your-plos-figures-graphics/"

Reviewers' comments:

Reviewer's Responses to Questions

**Comments to the Author**

1. Is the manuscript technically sound, and do the data support the conclusions?

Reviewer #1: Yes

Reviewer #2: Partly

2. Has the statistical analysis been performed appropriately and rigorously? 

Reviewer #1: No

Reviewer #2: I Don't Know

3. Have the authors made all data underlying the findings in their manuscript fully available?

Reviewer #1: Yes

Reviewer #2: Yes

4. Is the manuscript presented in an intelligible fashion and written in standard English?

Reviewer #1: Yes

Reviewer #2: Yes

5. Review Comments to the Author

Reviewer #1: The submitted manuscript is interesting for the approached subject. I have some concerns although, and before I may recommend the Editorial office a possible publication, I am asking the authors to clarify this.

Page 4, line 97: the authors stated "until constant weight". If the propolis extract is dried completely, how is this statement used? I believe the extract is completely dried and then weighted to make afterwards the dilutions requested for the study. Am I right? (maybe I misunderstood).

Page 14, line 294-295: prenylated phenyl propanoids and isoflavonoids are not detected in the studied samples? They are the main flavonoids in red and green propolis as stated also in the ISO 24381:2024 - Bee propolis — Specifications... The authors did not find them at all? These compounds have a huge bioactivity for green and red propolis. From the chromatograms presented as supplementary files, I could not figure out what were the identified compounds ...

Please clear this matter..

Reviewer #2: Analysis of chemical composition can be improved. I consider that there are some errors or omissions that must be addressed:

1. I don't see how the quantification process was done. UV detector or mass detector?

2. Mass spectrometer conditions are not declared

3. Did authors employ distilled water for ULPC-UV-MS analysis?

4. How the compounds were identified? Mass data? Only retention time? Propolis is a very complex mixture of compounds in which both isomers and isobars are common. Comparison based on retention times is not enough!!!!

5. I don’t see typical chemical constituents of both red and green Brazilian propolis in table 2

6. PLOS authors have the option to publish the peer review history of their article (what does this mean? ). If published, this will include your full peer review and any attached files.

**Do you want your identity to be public for this peer review?** For information about this choice, including consent withdrawal, please see our Privacy Policy .

Reviewer #1: **Yes: ** Otilia Bobis Senior researcher PhD

Reviewer #2: No

---

## [Author Response · Author response to Decision Letter 1]

31 Dec 2024

'Response to Reviewers'

Reviewers' comments

Has the statistical analysis been performed appropriately and rigorously?

Reviewer #1: No

Reviewer #2: I Don't Know

Response:The statistical analysis has been conducted appropriately and rigorously. In the methodology section, particularly in the description of the in-silico study, additional lines were included to enhance clarity and understanding. These added explanations can now be found in lines (146-148) of the manuscript.

Reviewer #1: The submitted manuscript is interesting for the approached subject. I have some concerns although, and before I may recommend the Editorial office a possible publication, I am asking the authors to clarify this. Page 4, line 97: the authors stated, "until constant weight". If the propolis extract is dried completely, how is this statement used? I believe the extract is completely dried and then weighted to make afterwards the dilutions requested for the study. Am I right? (maybe I misunderstood).

The authors would like to thank the reviewer for their observation. You are correct in your interpretation, and we apologize if the wording was not entirely clear. The phrase "until constant weight" refers to the drying process of the propolis extract until no further weight loss is observed, indicating that the extract has reached its stable weight. Indeed, after achieving constant weight, the extract is weighed and then diluted according to the study requirements. For further clarification, the method has been updated, now in lines (97-100).

Page 14, line 294-295: prenylated phenyl propanoids and isoflavonoids are not detected in the studied samples? They are the main flavonoids in red and green propolis as stated also in the ISO 24381:2024 - Bee propolis — Specifications... The authors did not find them at all? These compounds have a huge bioactivity for green and red propolis.

We appreciate your reference to ISO 24381:2024, which provides guidelines for the detection of chemical compounds in propolis. Following to this standard, our study identified several phenolic compounds, including phenylpropanoids or phenolic acids (caffeic acid, ferulic acid, and p-coumaric acid), flavonoids and derivates (Rutin, quercetin, naringenin, kaempferol) phenols (resorcinol, eugenol, vanillin), phenolic esters such as chlorogenic acid, and Gallic acid were identified, as detailed in (Table 2).

However, the absence of certain characteristic compounds, such as artipellin C (prenylated phenylpropanoids), formononetin (isoflavone), or pinocembrin, does not necessarily imply their non-existence in the analyzed samples. Our approach focused on identifying compounds that, based on the literature and a previous review published by our team (Silva-Beltrán et al., 2021), as well as our prior experience, were expected be present in both types of propolis (from Brazil and Mexico), given the diversity of ecosystems in each region. Therefore, we did not specifically target these markers but instead focused on compounds previously reported in propolis samples from both countries. In this regard, the validity of the in-silico study was ensured by using only properly characterized molecules, which guaranteed the reliability of the computational models employed.

To further address your concern, we have included an additional clarification in the discussion section, which can be found in lines [283-292]. We would also to mention that our research team plans to continue this study by incorporating other compounds considered markers for different types of propolis exploring a broader range of bioactive effects and mechanisms of action.

From the chromatograms presented as supplementary files, I could not figure out what were the identified compounds ...

Please clear this matter..

We apologize for any confusion. Compound identification was based on retention times and absorption spectra obtained during chromatographic analysis, using external standards. For clarity, we included legible figures to better highlight the compounds identified in the chromatograms.

S1 Fig. UPLC chromatogram of phenolic compound standards analyzed at 280 nm [y axis = intensity (absorbance unit, AU); x axis = retention time (min). Peaks: 1, gallic acid; 2, resorcinol; 3, chlorogenic acid; 4, caffeic acid; 5, vanillin; 6, coumaric acid; 7, ferulic acid; 8, rutin; 9, naringenin; 10, quercetin;11, kaempferol, and 12, eugenol.

S2 Fig. UPLC chromatogram of phenolic compounds present in G. Peaks: (3) naringenin, (6) coumaric acid, (10) quercetin, (11) kaempferol, (12) eugenol.

S3 Fig. UPLC chromatogram of phenolic compounds present in GS. (9) naringenin, (10) quercetin, (11) kaempferol, (12) eugenol.

S4 Fig. UPLC chromatogram of phenolic compounds present in RB. (3) naringenin, (4) caffeic acid, (5) vanillin, (6) coumaric acid, (7) ferulic acid, (9) naringenin, (10) quercetin, (11) kaempferol, (12) eugenol.

S5 Fig. UPLC chromatogram of phenolic compounds present in RA. (9) naringenin, (10) quercetin, (11) kaempferol, (12) eugenol.

References

Silva-Beltrán, N.P.; Umsza-Guez, M.A.; Ramos Rodrigues, D.M.; Gálvez-Ruiz, J.C.; de Paula Castro, T.L.; Balderrama-Carmona, A.P. Comparison of the Biological Potential and Chemical Composition of Brazilian and Mexican Propolis. Appl. Sci. 2021, 11, 11417. https://doi.org/10.3390/app112311417

Reviewer #2: Analysis of chemical composition can be improved. I consider that there are some errors or omissions that must be addressed:

Thank you for your comments. We appreciate your suggestions and have carefully reviewed your analysis. Below are the responses to each of them.

1. I don't see how the quantification process was done. UV detector or mass detector? Mass spectrometer conditions are not declared

We apologize for any confusion. To clarify, the quantification process was carried out using a UV detector, not a mass spectrometer. The quantification was based on retention times and absorption spectra obtained during chromatographic analysis, using external standards. The use of external standards allowed for accurate compound quantification through calibration curves. We have corrected the description of the protocol for greater clarity. Please refer to the updated details in line (126).

2. Did authors employ distilled water for ULPC-UV-MS analysis?

We employed HPLC-grade solvents, which is standard practice to ensure the quality and accuracy of the chromatographic analysis. For further clarification, the method has been updated. Please see now the update in line (129).

3. How the compounds were identified? Mass data? Only retention time? Propolis is a very complex mixture of compounds in which both isomers and isobars are common. Comparis on based on retention times is not enough!!!!

We appreciate your comment and understand the complexity of the propolis samples, which contain isomers and isobars. We would like to clarify that the identification of the compounds was not based solely on retention times, but on a combination of two methods:

Retention Times: External standards were used to establish retention times, as shown in Figure S1, allowing for an initial identification of the compounds.

Absorption Spectra: Identification was corroborated by comparing the absorption spectra obtained during chromatographic analysis, providing greater certainty. This protocol was previously validated by our team (Balderrama-Carmona et al., 2020), where reproducibility, linearity, and detection limits were reported, along with the absorption spectra of the identified compounds. For clarity, enlarged figures have been included in the supplementary files, in order to better highlight the compounds identified in the chromatograms.

Balderrama-Carmona, A.P.; Silva-Beltrán, N.P.; Gálvez-Ruiz, J.-C.; Ruíz-Cruz, S.; Chaidez-Quiroz, C.; Morán-Palacio, E.F. Antiviral, Antioxidant, and Antihemolytic Effect of Annona muricata L. Leaves Extracts. Plants 2020, 9, 1650. https://doi.org/10.3390/plants9121650

5. I don’t see typical chemical constituents of both red and green Brazilian propolis in table 2

We acknowledge that the absence of certain characteristic or typical compounds in Brazilian propolis, such as artipellin C, formononetin, does not necessarily imply their non-existence in the analyzed samples. Our approach focused on identifying compounds that, based on the literature and a previous review published by our team (Silva-Beltrán et al., 2021), as well as our prior experience, we knew would be present in both types of propolis (from Brazil and Mexico), given the diversity of ecosystems in each region. In this regard, the validity of the in-silico study was also ensured by maintaining a more rigorous and precise approach, using only properly characterized molecules, which ensured the reliability of the computational models used. However, to address your concern, we have decided to include an additional clarification in the discussion section, which can be found in line [283-292].

Furthermore, we would like to mention that our research team plans to continue this study by incorporating other compounds considered markers or typical chemical constituents for different types of propolis, to explore a broader range of bioactive effects and mechanisms of action.

Silva-Beltrán, N.P.; Umsza-Guez, M.A.; Ramos Rodrigues, D.M.; Gálvez-Ruiz, J.C.; de Paula Castro, T.L.; Balderrama-Carmona, A.P. Comparison of the Biological Potential and Chemical Composition of Brazilian and Mexican Propolis. Appl. Sci. 2021, 11, 11417. https://doi.org/10.3390/app112311417

---

## [Decision Letter · Decision Letter 1]

4 Feb 2025

PONE-D-24-54321R1Brazilian and Mexican Propolis and their possible Mechanism of Action against non-enveloped VirusesPLOS ONE

Dear Dr. Silva Beltrán,

Thank you for submitting your manuscript to PLOS ONE. After careful consideration, we feel that it has merit but does not fully meet PLOS ONE’s publication criteria as it currently stands. Therefore, we invite you to submit a revised version of the manuscript that addresses the points raised during the review process.

We look forward to receiving your revised manuscript.

Kind regards,

José M. Alvarez-Suarez

Academic Editor

PLOS ONE

Journal Requirements:

Reviewers' comments:

Reviewer's Responses to Questions

**Comments to the Author**

1. If the authors have adequately addressed your comments raised in a previous round of review and you feel that this manuscript is now acceptable for publication, you may indicate that here to bypass the “Comments to the Author” section, enter your conflict of interest statement in the “Confidential to Editor” section, and submit your "Accept" recommendation.

Reviewer #1: All comments have been addressed

Reviewer #2: All comments have been addressed

Reviewer #3: (No Response)

2. Is the manuscript technically sound, and do the data support the conclusions?

Reviewer #1: Yes

Reviewer #2: Partly

Reviewer #3: Yes

3. Has the statistical analysis been performed appropriately and rigorously? 

Reviewer #1: Yes

Reviewer #2: I Don't Know

Reviewer #3: Yes

4. Have the authors made all data underlying the findings in their manuscript fully available?

Reviewer #1: Yes

Reviewer #2: Yes

Reviewer #3: No

5. Is the manuscript presented in an intelligible fashion and written in standard English?

Reviewer #1: Yes

Reviewer #2: Yes

Reviewer #3: Yes

6. Review Comments to the Author

Reviewer #1: I am satisfied with the answers that the authors gave to my concerns. For this reason I will suggest the editorial office to take into consideration for publication the study you have been conducted

Reviewer #2: I consider that the retention time and the comparison with the standards are not sufficient to unequivocally determine the structure of a compound in a product as complex as propolis. Nowadays, characterization by mass spectrometry is common and even several multiple reactions monitoring transitions are required to identify a compound.

The current requirements of validation processes for HPLC-MS constitute evidence of the rigor of structural characterization. Otherwise, there is a risk of errors in structural identification.

On line 125 the phrase ¨distilled water¨ still remains.

Reviewer #3: The manuscript needs some corrections to improve its quality. I attached a file containing my comments

7. PLOS authors have the option to publish the peer review history of their article (what does this mean? ). If published, this will include your full peer review and any attached files.

**Do you want your identity to be public for this peer review?** For information about this choice, including consent withdrawal, please see our Privacy Policy .

Reviewer #1: **Yes: ** Bobis otilia, PhD

Reviewer #2: No

Reviewer #3: **Yes: ** Mohammed Elimam Ahamed Mohammed

---

## [Author Response · Author response to Decision Letter 2]

11 Feb 2025

Reviewer #1: I am satisfied with the answers that the authors gave to my concerns. For this reason I will suggest the editorial office to take into consideration for publication the study you have been conducted

Thank you for your positive feedback and for taking the time to review our manuscript. We truly appreciate your thoughtful comments and your support in considering our study for publication.

Reviewer #2:

On line 125 the phrase ¨distilled water¨ still remains

We have reviewed now is line 140.

I consider that the retention time and the comparison with the standards are not sufficient to unequivocally determine the structure of a compound in a product as complex as propolis. Nowadays, characterization by mass spectrometry is common and even several multiple reactions monitoring transitions are required to identify a compound.

The current requirements of validation processes for HPLC-MS constitute evidence of the rigor of structural characterization. Otherwise, there is a risk of errors in structural identification.

We sincerely appreciate your valuable comments and your interest in enhancing the quality of our work. We fully acknowledge your concerns regarding the unequivocal identification of compounds in a complex matrix such as propolis. However, we would like to emphasize that our methodological approach is based on validated techniques, which have been extensively reported in the literature for the identification and quantification of compounds in complex natural products like propolis. For instance, Devequi et al. (1) successfully used HPLC-UV for the analysis of propolis, and Zhang et al. (2023) (2) demonstrated that both HPLC-UV and HPLC-MS/MS can be effectively used for the analysis of propolis when good quality control conditions.

Our analytical protocol has been rigorously validated in terms of reproducibility (RSD < 5%), linearity (R² > 0.99), and detection limits, ensuring the reliability of the reported results. The information was added (see Lines 134-138; 168-178) and S1 Table.

Additionally, the compounds identified were selected based on their previously documented occurrence in propolis from Brazil and Mexico (1–6), minimizing the risk of misidentification.

Considering these points, we are confident that the data obtained remain scientifically robust and provide relevant insights into the composition of the analyzed samples. We have also made sure to transparently describe our methodology, providing sufficient detail for readers to understand both the scope and the limitations of our approach.

We recognize that techniques such as mass spectrometry would add significant value to future studies. Therefore, we will take your recommendation into careful consideration for the next stage of our study, which will focus on working with propolis markers. To this end, we plan to conduct the research using HPLC-MS.

Once again, we sincerely appreciate your constructive feedback, which will undoubtedly contribute to the ongoing improvement of our research.

Lines 134-138

The phenolic compounds were quantified using external standard curves prepared from pure standards. For compound identification, the absorption spectrum generated by integrating the area under the curve at the retention time detected in the standards was compared. Quality parameters were reproducibility, linearity, and limit of quantification (LOQ).

Lines:168-178

A phenolic profile was performed on samples G, GS, RA, and RB using 12 standards. Standard curves for the phenolic compounds are shown in S2 figure, while chromatograms and detection peaks for G, GS, RB, and RA are provided in the supplementary material (S3, S4, S5, and S6 Figs). Quality control parameters included linearity (R² > 0.99) and precision (RSD < 4.5%), as detailed in S1 Table. Linearity was evaluated by analyzing different standard solutions, and calibration curves were constructed by plotting the under-curve area versus concentration. The curves exhibited slopes of 86.136, intercepts of -105.74, and correlation coefficients of 0.99, with a limit of quantification (LOQ) of 0.2 μg/mL. A total of 10 constituents were identified, and Table 2 shows the concentration of the phenolic components detected in the red propolis from Brazil, as well as in the green propolis from Brazil and Mexico (Table 2).

References

1. Devequi-Nunes D, Machado BAS, Barreto GdA, Rebouças Silva J, da Silva DF, da Rocha JLC, et al. Chemical characterization and biological activity of six different extracts of propolis through conventional methods and supercritical extraction. PLoS One. 2018;13(12):e0207676

2. Zhang Y, Cao C, Yang Z, Jia G, Liu X, Li X, et al. Simultaneous determination of 20 phenolic compounds in propolis by HPLC-UV and HPLC-MS/MS. Journal of Food Composition and Analysis. 2023;115:104877

3. Hernandez J, Goycoolea FM, Quintero J, Acosta A, Castañeda M, Dominguez Z, et al. Sonoran propolis: chemical composition and antiproliferative activity on cancer cell lines. Planta medica. 2007;73(14):1469-74.

4. Silva-Beltrán NP, Umsza-Guez MA, Ramos Rodrigues DM, Gálvez-Ruiz JC, de Paula Castro TL, Balderrama-Carmona AP. Comparison of the biological potential and chemical composition of Brazilian and Mexican propolis. Applied Sciences. 2021;11(23):11417.

5. Devequi-Nunes D, Machado BAS, Barreto GdA, Rebouças Silva J, da Silva DF, da Rocha JLC, et al. Chemical characterization and biological activity of six different extracts of propolis through conventional methods and supercritical extraction. PLoS One. 2018;13(12):e0207676.

6. Salatino A, Salatino MLF, Negri G. How diverse is the chemistry and plant origin of Brazilian propolis? Apidologie. 2021:1-23.

Reviewer #3: The manuscript needs some corrections to improve its quality. I attached a file containing my comments

The title

Mechanism of action of an antibiotic is not just its binding to a target protein, It involves the subsequent actions on specific processes such as translation (protein synthesis) transcription (RNA synthesis), replication (DNA synthesis) and reverse transcription (Synthesis of DNA from RNA) [1-3]. I suggest the following title:

Brazilian and Mexican Propolis and their Invitro and Insilco activities against nonenveloped Viruses

We appreciate your comments and, in line with your observations, we have had the opportunity to improve our results and identify more relevant data derived from the mechanisms of action, which led us to decide not to change the title. With the intention of strengthening it (title), we have made improvements to the abstract (30-32; 34-37), discussion (361-374), and conclusion (380-383), where we have added information related to the mechanism of action.

Discussion

Lines 361-374

Furthermore, Figure 1A shows that phenolic compounds exhibit greater affinity for DNA bacteriophages. This finding is consistent with the in vitro study (Table 3), which indicates that the DNA bacteriophage ΦX174 is more sensitive.

To link the in vitro effects with the known functions of specific viral proteins, we selected four proteins involved in capsid structure, genome binding, and infection mechanisms. Specifically, we analyzed MS2 (PDB ID: 1AQ3) and ΦX174 (PDB ID: 2BPA) both of which are required for genome attachment-dependent capsid assembly.

To determine the effect on viral processes, we selected spike proteins, including 5TC1 (chain M) for MS2 and 1CD3 (chain G) for ΦX174. The former is responsible for virus attachment to the F pilus and genome delivery, while the latter is associated with effective infection (1)

Therefore, the antiviral properties of propolis G and GS observed in this study could be attributed to their higher content and composition of phenolic compounds, which enhance synergistic (protein-ligand) effects, facilitating interactions that neutralize the capsid proteins required for genome attachment-dependent capsid assembly of both non-enveloped viruses.

Conclusion

Lines 380-383

Altogether, our results highlight the effect of propolis on phage infection and suggest a possible mechanism of action through binding to capsid proteins, which could prevent the proper formation of the capsid-genome complex and block viral infection.

Abstract

1- Rephrase the abstract to reflect the suggested title.

The information was added Lines: 30,32.

In silico molecular docking was also conducted to determine binding energy and molecular interaction and putative mechanism of propolis phenolic compounds with two viral capsid proteins and two proteins involved in viral replication and infection.

The information was added Lines: 34-37

Molecular docking simulations revealed that ΦX174 was also more sensitive to the phenolic compounds and that the combination of quercetin and kaempferol showed the greatest antiviral effect as a possible mechanism, through binding to the viral capsid proteins near the viral genome binding sites.

2- you stated (Propolis samples were characterized by ultra-performance liquid chromatography (UPLC) before testing), I suggest to rephrase it stating that you measured the concentration of 12 phenolic compounds using the UPLC.

The information was added: (lines 26-28)

Propolis samples were characterized by performing a phenolic profile using ultra-performance liquid chromatography (UPLC), which included 12 phenolic compounds such as phenylpropanoids, flavonoids, phenols, and phenolic acids

Introduction

I suggest to add a paragraph about the non-envelope viruses; their types, severity and structure

The information was added: (lines 65-71)

Several studies have employed bacteriophage models to investigate the in vitro virucidal effects of natural substances �4, 12�. Among them, the bacteriophages MS2 (single-stranded RNA) and ΦX174 (single-stranded DNA), both non-enveloped virus, have been employed as viral models. Non-enveloped viruses are typically more resistant to adverse environmental conditions and to the action of antimicrobials �13�. The MS2 bacteriophage has been used as an experimental surrogate for SARS-CoV-2 �14�, while ΦX174 used as an enteric virus surrogate �15

Material and methods:

Please describe in details how you measured the 12 phenolic acids How you created the standard curves and what are the quality parameters you used to control your assays.

The information was added: (lines 133-140)

The phenolic compounds were quantified using external standard curves prepared from pure standards, with concentrations ranging from 0.1 to 100 μg/mL. For compound identification, the absorption spectrum generated by integrating the area under the curve at the retention time detected in the standards was compared. Quality parameters were reproducibility, linearity, and limit of quantification (LOQ)

Results

Add the standard curves of the phenolic acids including the straight-line equation, R2 and r values.

The information was added: (lines 168-178)

Phenolic composition of propolis

A phenolic profile was performed on samples G, GS, RA, and RB using 12 standards. Standard curves for the phenolic compounds are shown in S2 figure, while chromatograms and detection peaks for G, GS, RB, and RA are provided in the supplementary material (S3, S4, S5, and S6 Figs). Quality control parameters included linearity (R² > 0.99) and precision (RSD < 4.5%), as detailed in S1 Table. Linearity was evaluated by analyzing different standard solutions, and calibration curves were constructed by plotting the under-curve area versus concentration. The curves exhibited slopes of 86.136, intercepts of -105.74, and correlation coefficients of 0.99, with a limit of quantification (LOQ) of 0.2 μg/mL.

A total of 10 constituents were identified, and Table 2 shows the concentration of the phenolic components detected in the red propolis from Brazil, as well as in the green propolis from Brazil and Mexico (Table 2).

Discussion

Discuss the results of the 12 phenolic acids you investigated

The requested information has now been integrated, and you can view it in lines

Lines 292-299

Propolis extracts from Mexico and Brazil were analyzed to evaluate their antiviral activities in vitro and explore possible mechanisms using of action molecular docking on the detected chemical constituents. A targeted phenolic profile was performed on samples G, GS, RA, and RB, including phenylpropanoids or phenolic acids (caffeic acid, ferulic acid, and p-coumaric acid), flavonoids and their derivatives (rutin, quercetin, naringenin, kaempferol), phenols (resorcinol, eugenol, vanillin), and phenolic esters such as chlorogenic acid and gallic acid, as detailed in Table 2. The presence of these compounds was corroborated through chromatographic analysis.

Lines 306-323

However, this phenylpropanoid was only detected in GS. Brazilian green propolis is mostly composed of prenylated phenylpropanoids, mainly artepillin C; Brazilian red propolis mainly contains isoflavonoids (biochanins, medicarpin) �24�. Although we acknowledge the potential presence of other characteristic compounds in the studied propolis samples, this study focused on identifying compounds that, based on previous research conducted by our team, we knew could be detected in both Brazilian and Mexican propolis�9�. This approach ensured the validity of the docking analysis. Consequently, were also detected in Brazilian propolis compounds, such as chlorogenic acid, caffeic acid, vanillin, coumaric acid, ferulic acid, naringenin, kaempferol, eugenol and quercetin �9�. Quercetin, kaempferol and eugenol were detected in all propolis samples G, GS, RB, RA. In addition to the compound identified, it was observed that red propolis presents significant concentrations of kaempferol with 45.752, 73.86 and 38.47 mg/g for G, RB, RA respectively. However, rutin (quercetin 3-O-rutinoside) was not detected in all propolis samples, likely because the flavonoids in propolis are mostly aglycons, making rutin a rare major component. Another compound that was not detected was resorcinol, which was expected to be found in Brazilian propolis, particularly in samples from hives near mangroves in the states of Bahía (RB) and Alagoas (RA) (29). The absence of specific compounds in propolis depends on the botanical diversity of the region.

Lines: 361-363

Furthermore, Figure 1A shows that phenolic compounds exhibit greater affinity for DNA bacteriophages. This finding is consistent with the in vitro study (Table 3), which indicates that the DNA bacteriophage ΦX174 is more sensitive.

Lines 368-379

To link the in vitro effects with the known functions of specific viral proteins, we selected four proteins involved in capsid structure, genome binding, and infection mechanisms. Specifically, we analyzed MS2 (PDB ID: 1AQ3) and ΦX174 (PDB ID: 2BPA) both of which are required for genome attachment-dependent capsid assembly.

To determine the effect on viral processes, we selected spike proteins, including 5TC1 (chain M) for MS2 and 1CD3 (chain G) for ΦX174. The former is responsible for virus attachment to the F pilus and genome delivery, while the latter is associated with effective infection (1)

Therefore, the antiviral properties of propolis G and GS observed in this study could be attributed to their higher content and composition of phenolic compounds, which enhance synergistic (protein-ligand) effects, facilitating interactions that neutralize the capsid proteins required for genome attachment-dependent capsid assembly of both non-enveloped viruses.

---

## [Decision Letter · Decision Letter 2]

2 Apr 2025

Brazilian and Mexican Propolis and Their possible Mechanism of Action Against Non-Enveloped Viruses

PONE-D-24-54321R2

Dear Dr. Silva Beltrán,

We’re pleased to inform you that your manuscript has been judged scientifically suitable for publication and will be formally accepted for publication once it meets all outstanding technical requirements.

Kind regards,

José M. Alvarez-Suarez

Academic Editor

PLOS ONE

Additional Editor Comments:

As as an editor and expert in the field, I have carefully reviewed the manuscript along with the corrections and modifications made by the authors in response to the reviewers' comments, particularly those raised by Reviewer 2. I acknowledge Reviewer 2’s concern regarding the use of more sensitive methods that could allow for a more thorough characterization of the matrix. However, I consider that the methodology employed by the authors is appropriate and sufficient to address the research questions posed in this study.

That said, I also recognize the relevance and value of the more sensitive techniques suggested by the reviewer, as they indeed offer a deeper and more detailed analysis. Nevertheless, I maintain the view that the methods used by the authors are valid and do not compromise the reliability or integrity of the results.

Reviewers' comments:

Reviewer's Responses to Questions

**Comments to the Author**

1. If the authors have adequately addressed your comments raised in a previous round of review and you feel that this manuscript is now acceptable for publication, you may indicate that here to bypass the “Comments to the Author” section, enter your conflict of interest statement in the “Confidential to Editor” section, and submit your "Accept" recommendation.

Reviewer #2: All comments have been addressed

Reviewer #3: All comments have been addressed

2. Is the manuscript technically sound, and do the data support the conclusions?

Reviewer #2: Partly

Reviewer #3: Yes

3. Has the statistical analysis been performed appropriately and rigorously? 

Reviewer #2: I Don't Know

Reviewer #3: Yes

4. Have the authors made all data underlying the findings in their manuscript fully available?

Reviewer #2: Yes

Reviewer #3: Yes

5. Is the manuscript presented in an intelligible fashion and written in standard English?

Reviewer #2: Yes

Reviewer #3: Yes

6. Review Comments to the Author

Reviewer #2: My practical experience tells me that coelution is a very common phenomenon in propolis chromatograms. Therefore, using UV spectroscopy for quantification involves a significant error. It is clear from the propolis chromatograms that coelution occurs, and the chromatographic resolution is far from optimal for proper quantification. For these reasons, detection and quantification by mass spectrometry is necessary.

Page 1, lines 28-29

The authors indicate:

…Quercetin, eugenol, kaempferol and naringenin were the most abundant compounds found in propolis.

The main constituents of Brazilian red propolis are isoflavans, isoflavones and pterocarpans (None were detected in this study). Besides, the main constituents of Brazilian green propolis are artepillicin C, drupanin, p-coumaric acid (unique product detected) and hydrocinnamic acid (Bankova et al., 2019). On the other hand, many intense chromatographic peaks (Figure S3, S4, S5) were not identified. Most likely, these peaks were originated by the typical constituents mentioned above. Therefore, some major constituents of propolis were ignored during the study.

The doubt regarding the influence of the main chemical components of both green and red propolis on the experiments carried out remains unanswered. From my point of view, the focus should have been on the main chemical components of each propolis and not on those that could be detected.

Bankova, V., Bertelli, D., Borba, R., Conti, B. J., da Silva Cunha, I. B., Danert, C., & Zampini, C. (2019). Standard methods for Apis mellifera propolis research. Journal of Apicultural Research, 58(2), 1-49.

Reviewer #3: Thank you very much for your positive responses to the majority of my comments. Regarding the title of your article, I accept your defence. All the best

7. PLOS authors have the option to publish the peer review history of their article (what does this mean? ). If published, this will include your full peer review and any attached files.

**Do you want your identity to be public for this peer review?** For information about this choice, including consent withdrawal, please see our Privacy Policy .

Reviewer #2: No

Reviewer #3: **Yes: ** Mohammed Elimam

---

## [Editor Report · Acceptance letter]

PONE-D-24-54321R2

PLOS ONE

Dear Dr. Silva Beltrán,

I'm pleased to inform you that your manuscript has been deemed suitable for publication in PLOS ONE. Congratulations! Your manuscript is now being handed over to our production team.

Kind regards,

on behalf of

Professor José M. Alvarez-Suarez

Academic Editor

PLOS ONE